# Influence of Particle Morphology of Ground Fly Ash on the Fluidity and Strength of Cement Paste

**DOI:** 10.3390/ma14020283

**Published:** 2021-01-07

**Authors:** Juntao Ma, Daguang Wang, Shunbo Zhao, Ping Duan, Shangtong Yang

**Affiliations:** 1International Joint Research Lab for Eco-Building Materials and Engineering of Henan, North China University of Water Resources and Electric Power, Zhengzhou 450045, China; wang18838970093@163.com (D.W.); sbzhao@ncwu.edu.cn (S.Z.); 2School of Civil Engineering and Communication, North China University of Water Resources and Electric Power, Zhengzhou 450045, China; 3Department of Civil and Environmental Engineering, University of Strathclyde, Glasgow G1 1XJ, UK; shangtong.yang@strath.ac.uk; 4Key Laboratory of Geological Survey and Evaluation of Ministry of Education, Faculty of Materials Science and Chemistry, China University of Geosciences, Wuhan 430074, China; duanping@cug.edu.cn; 5Zhejiang Institute, China University of Geosciences (Wuhan), Hangzhou 311305, China; 6Guangxi Key Laboratory of New Energy and Building Energy Saving, Guilin University of Technology, Guangxi 541004, China

**Keywords:** ground fly ash, spherical destruction, cement paste, fluidity, strength

## Abstract

The grinding process has become widely used to improve the fineness and performance of fly ash. However, most studies focus on the particle size distribution of ground fly ash, while the particle morphology is also an important factor to affect the performance of cement paste. This article aims at three different kinds of ground fly ash from the ball mill and vertical mill, and the particle morphology is observed by scanning electron microscopy (SEM) to calculate the spherical destruction (the ratio of spherical particles broken into irregular particles in the grinding process of fly ash), which provides a quantification of the morphology change in the grinding process. The fluidity of cement paste and the strength of cement mortar are tested to study the relation of spherical destruction and fluidity and strength. The results show that the spherical destruction of ground fly ash in a ball mill is more than 80% and that in a vertical mill with a separation system is only 11.9%. Spherical destruction shows a significant relation with the fluidity. To different addition of ground fly ash, the fluidity of cement paste decreases with the increase of spherical destruction. To the strength of cement paste, particle size distribution and spherical destruction are both the key factors. Therefore, spherical destruction is an important measurement index to evaluate the grinding effect of the fly ash mill.

## 1. Introduction

Fly ash has been one of the first artificial additions used in the production of cement and concrete since the first decades of the 20th century [1,2,3]. It is well known that the addition of fly ash in cement or concrete can improve some performances of construction materials and decrease the consumption of cement in the construction project [4,5], which contributes to the low-carbon economy in concrete industry. Fly ash consists of a large proportion of active SiO_2_ and Al_2_O_3_, and the particles of fly ash are spherical in shape. In the previous studies of fly ash, the pozzolanic effect [6,7,8] and filling effect [9] improve the strength of concrete, while the ball-bearing effect [10,11] enhances the [12] fluidity of concrete. Therefore, besides the chemical composition, particle size distribution and particle morphology are the main factors to affect the properties of fly ash.

For fly ash, the relation between particle size distribution and properties has been studied a lot. In most of the power plants in China, conforming to Chinese National Standard GB/T 1596-2017, fly ash is classified into different grades by the fineness. The finer fly ash is much easier to be applied in the structure. The classified ultrafine fly ash (UFFA) is widely used in concrete to enhance the properties [13,14,15,16,17,18], as it is proven that moderate UFFA addition improves the compressive strength at all ages [19,20]. Meanwhile, the utilization of coarser fly ash is limited in the high-performance concrete construction. Many researches [5,21,22,23,24] showed that high-volume or low-grade fly ash decreases the early strength of concrete. However, the supply of ultrafine fly ash is obviously behind the demand for high-performance concrete, while the coarser fly ash is overabundant and difficult to be disposed. To achieve the fineness demand of fly ash in concrete project, grinding fly ash by the mill is one of the popular approaches to reduce the particle size and improve the performance of concrete containing fly ash. In recent years, a lot of experiments are carried to study the influence of particle size distribution on the fluidity and strength of concrete. Krishnaraj [25], Zhao [26], and Duan [27] used different mills to produce ground fly ash, which indicated that the grinding process improved the fineness and activity effectively.

Meanwhile, the particle morphology is another main influence factor to the properties of fly ash. In the grinding process, some spherical particles, especially the unburned carbon particles and hollow particles, are easily crushed and the cracking of cenospheres and plerospheres may produce some shell-shaped fragments [28]. The morphology of fly ash particles may be remarkably different after the grinding process [29,30]. Lanzerstorfer has compared the grinded fly ash and classified fly ash, which shows that the shape changes from round to angular during milling, while classification does not affect the particle shape [30].

The relation of particle morphology of fly ash and cement paste properties, including the fluidity and strength, is studied widely. It is generally accepted that the spherical shape morphology and vitreous character of fly ash provides the lubricant effect [31] and ball-bearing effect [11,32,33,34], which improve the fluidity of concrete effectively [35,36,37,38,39,40]. In these studies, spherical particles in fly ash, reduce the internal friction and lubricate the surface of cement particles, which is similar with the effect of balls. Therefore, the morphology change in the grinding process may influence the effects on the fluidity. Li [41] and Li [42] studied the fluidity of cement paste containing ground fly ash of different particle size distribution. Meanwhile, two different types of fly ash, from grinding and air classification, were used to test the effect to relative workability factor, which showed that the relative workability factor value is greater when non-destructive methods were used [29,30]. The spherical particle destruction also leads to the activity change of fly ash after the grinding process. The studies of Young [8] indicate that the pozzolanic reactivity of fly ash is greatly affected by the amorphous phase content. Han [43] compared the activity of raw fly ash and ground fly ash, while cement mortar and concrete strength is tested. Paya [44,45] calculated the Compressive strength gain and pozzolanic effectiveness ratio, which is greater when the ground fly ash is added. Lanzerstorfer [30] and Jones [46] evaluate different addition of ultrafine fly ash and test the strength of concrete. Itskos [47] indicates that the fly ash in the particle size range of 75–150 μm obtains better pozzolanic effect, and the grinding process crumble the superficial glass and increase the specific surface area.

However, most of the research related to the particle morphology focuses on the rheological mechanism and hydration mechanism of ground fly ash in cement paste. If we want to establish a certain relation to evaluate the influence of particle morphology of fly ash, it is necessary to give a quantification to the spherical destruction of fly ash, which is difficult to achieve by traditional measurement. Meanwhile, image analysis is used to test the performance index of powder particles in some studies. Wang [48] used image analysis coupled with backscattered electron (BSE) to evaluate the special surface area, which is able to provide a satisfying estimation. Image analysis is also used to study the pore structure of cement paste [49], reaction degree [50], glass phases [51] and elongation [52] of fly ash. Scanning electron microscopy (SEM) is always used to observe the morphology of fly ash particles. If sufficient images are provided, the proportion of spherical particles in the whole ground fly ash particles, can give a new way to evaluate the influence of fly ash morphology on the fluidity and strength of cement paste containing ground fly ash. In this paper, fly ash is grinded in different grinding systems and the particle size is tested. The spherical destruction is observed by SEM, calculated and analyzed by Image-Pro Plus (Image-Pro Plus 6.0, Media Cybernetics, Rockville, MD, USA). The fluidity of cement paste and compressive strength of cement mortar are tested to study the relation of spherical destruction and fluidity and strength. The pore structure analysis and hydration product observation are also carried out to explore the mechanism.

## 2. Experimental

### 2.1. Grinding Conditions of Fly Ash

P.O 42.5 Cement and fly ash are used in this experiment. The fly ash is from the Datang power plant in Henan Province, Sanmeixia, China. The chemical composition of cement and fly ash is listed in Table 1. Raw fly ash is grinded in different grinding systems, including ball mill and vertical mill. The abrasive media of ball mill (SM-500, Wuxi building material instrument and machinery Factory, Wuxi, Jiangsu Province, China) is composed of steel balls and sections, of which the rotate speed is 48 rpm and the media loading is 50%. The vertical mill system, assembled in Xi’an University of Architecture and Technology, contains a roller mill and pneumatic powder separation system, in which the fineness qualified ground fly ash is separated and the unqualified powder is returned to the mill. Three different grinding conditions are listed in Table 2.

### 2.2. Particle Size Distribution and Special Surface Area Analysis

Four fly ash samples are dispersed by laser and the particle size is analyzed by LS13320 laser particle size analyzer (Beckman Coulter, Indianapolis, IN, USA), as shown in Figure 1. The medium particle size (D50) is considered the average particle size, as shown in Table 3. The special surface area is tested by the Blaine method (BMY-6 tester) (Beijing Samyon Instruments, Beijing, China), as shown in Table 3.

From the analysis of raw fly ash, the size of about 82% fly ash particles is concentrated between 50 μm and 200 μm. After grinded in the ball mill, the main particle size interval moves to 0 μm to 50 μm, while the size of most vertical mill grinding fly ash particles is concentrated between 0 μm to 25 μm. The average particle size analysis, sample BF1 and BF2 in ball mill show about 20% size of raw fly ash, while sample VF shows about 12%. Meanwhile, the special surface area of BF2 is about 2 times that of raw fly ash and VF is nearly 2.2 times.

The fineness results show the difference of ground fly ash in ball mill and vertical mill. The separation system can separate the qualified particles and improve the grinding efficiency, which leads to the finer fly ash particles in a vertical mill than in the ball mill.

### 2.3. Micro Morphology Observation of Fly Ash

The morphology and spherical destruction of different fly ash (RF, BF1, BF2 and VF) are analyzed by the scan electron microscope. Then, fifty images are chosen in every sample to calculate the area of spherical particles and non-spherical particles by Image-Pro plus. The calculation method is listed in 3.2.

### 2.4. Fluidity Test of Cement Paste

The rheological property of cement paste containing different fly ash is evaluated by the fluidity of cement paste. The fluidity is determined conforming to Chinese National Standard GB/T 8077-2012, in which the fluidity test mold (top diameter: 36 mm; bottom diameter: 60 mm; height: 60 mm) and Vernier caliper are used. The expansion diameter is measured to evaluate the fluidity of cement paste. To each test 300 g cement or fly ash are used. Three groups of cement paste are tested in this experiment and the average value is calculated.

### 2.5. Strength Test of Hardened Cement Paste

The reaction degree is tested to evaluate the activity of fly ash, which refers to Chinese National Standard GB/T 12960-2007. The insoluble residue after hydrochloric titration is tested after curing for 1 d, 3 d, 7 d, 14 d, 21 d, 28 d and 90 d. The reaction degree is calculated according to Equation (1):(1)$$P_{f}\;=\;\left\{1-\frac{(1+w/b)\left[m_{un}-\left(1-\alpha \right)m_{0}\right]}{100\alpha m}\right\}\times 100\%$$
where *P****_f_*** is the reaction degree of fly ash (%); *m_un_* is the mass of insoluble residue (g); *α* is the substitution proportion (%); *m**_0_*** is the mass of insoluble residue when no fly ash added (g); *m* is the mass of cement paste sample (g); *w/b* is water to binder ratio.

The compressive strength of cement mortar containing different ground fly ash is tested after curing for 3 d, 7 d and 28 d, according to the Chinese National Standard GB/T 17671–1999. The curing temperature is 20 ± 2 °C and the relative humidity is no lower than 90%. Three groups of mortar are tested in this experiment and the average value is calculated. The loading speed of compressive test is 2400 N/s.

### 2.6. Micro Mechanism Analysis

In order to compare the microstructures of hydration products containing different types of fly ash, cement paste samples were prepared to analyze the pore distribution. RF, BF2, and VF are used in cement paste and a 100% cement paste sample is also prepared. The fly ash addition is 30% and the W/B ratio is 0.4. Cement paste samples are curing in the deionized water for 28 d. The hydration of the cement paste is stopped by immersing the samples into isopropyl alcohol for 24 h followed by drying in an oven at 6 °C for 24 h. Mercury intrusion porosimeter (MIP, Malvern, UK) is used to test the pore structure, and scan electron microscope (SEM, JSM-6610LV, Hongkong, China) is used to observe the micro structure of hardened cement paste.

## 3. Spherical Destruction Analysis of Fly Ash

### 3.1. SEM Images

Scan electron microscope is used to analyze the morphology and spherical destruction of different fly ash. The SEM images are listed in Figure 2. Most of the particles in raw fly ash is not angular, mainly spherical, generated in a high temperature of furnace. However, after grinded in the mill, a large proportion of fly ash particles are broken to angular particles. In the study of Paya [28], most of the particles remained unaltered, while some spheres are broken down to shell-shaped fragments (from cracking of cenospheres) and solid pieces (from solid particles), especially the brittle unburned carbon particles and hollow particles are easily crushed.

From the comparison of different images, the quantity of angular particles shows larger proportion in samples grinded in ball mill than in vertical mill. The angular particles are generated by the impact of abrasive media in the mill. The enclosed crushing space of the ball mill leads to the excessive grinding of some qualified fly ash particles, which are more probable to translate to angular particles. Meanwhile, the separation system of vertical mill guarantees the qualified particles from crushing, which protects more spherical particles from impact and destruction. The studies of Lanzerstorfer [30] and Itskos [47] indicated the shape transformation from round to angular in the grinding process, and classification contributes to the protection of fly ash particles.

### 3.2. Particle Area Analysis

Based on the difference between spherical particles and non-spherical particles, the area of different particles is calculated to determine the proportion of spherical destruction in fly ash grinding process, which is to analyze the relationship among the spherical destruction and activity and rheological property of fly ash in cement paste. Image-Pro plus is used to calculate the area of different particles. In the SEM observation process, for each kind of fly ash, one sample is chosen to represent the whole morphology. Five random areas are chosen for analyzation and 10 microscopic images are taken in every area. The selected 50 images are analyzed in Image-Pro plus as follows: Spherical particles are selected in the image and painted by a different color (Figure 3a,b), and the gray scale in Figure 3b of spherical particles, non-spherical particles and interspace is different; Interspace area is chosen and calculated in Figure 3c and spherical particles in Figure 3d, while the residual is the area of non-spherical particles. The average area of 50 images is calculated to determine the spherical particle proportion of sample RF, BF1, BF2, and VF.

From the analysis and calculation, the accumulation area of different fly ash is listed in Figure 4. The X axis indicates the 50 images of fly ash samples. There are spherical particles, non-spherical particles and interspace in every picture. The comparison of different samples shows that the spherical particle proportion in raw fly ash and samples ground in a vertical mill is higher than samples ground in a ball mill.

The analysis in Figure 3 shows that all particles in raw fly ash are formed at high temperature [1,31] and have not been destroyed. The non-spherical and spherical particles are both glassy spheres and spongy aggregates, while the shape of non-spherical particles are not so regular. Considering the complexity to calculate the destruction of these irregular particles, only the shape change of spherical particles is calculated in this analysis to measure the spherical removal of ground fly ash, which can reflect the destruction degree of particle morphology.

Spherical destruction in this experiment is defined as the comparison of spherical particle proportion in ground fly ash and raw fly ash. From the calculation of raw fly ash, the average proportion of spherical particles area in the total image area is 19.53%. From the analysis of three different kinds of ground fly ash, the spherical destruction is calculated by Equation (2):
(2)$$\mu =\frac{\alpha_{\mathrm{avgR}}-\alpha_{\mathrm{avgG}}}{\alpha_{\mathrm{avgR}}}\times 100\%$$
where μ is the spherical destruction; α_avgR_ is the average spherical particle proportion in raw fly ash; α_avgG_ is the average spherical particle proportion in ground fly ash.

The spherical destruction calculation of different fly ash is listed in Table 4.

The calculation shows that about 80% spherical particles are destroyed to non-spherical particles after grinding in ball mill for 1 h, and nearly 90% are destroyed after grinding for 2 h. Meanwhile, most spherical particles are protected from destroying in vertical mill grinding process and only 11.9% of spherical particles are destroyed.

## 4. Fluidity of Cement Paste Containing Fly Ash

### 4.1. Fluidity of Cement Paste

Different kinds of fly ash (RF, BF1, BF2, and VF) are used in cement paste and the addition is 10%, 20%, 30%, and 40% to replace cement, while cement paste containing no fly ash is also prepared and tested as a comparison sample. The water to binder ratio is 0.5. The fluidity of different cement paste is tested, and the results are shown in Figure 5.

From the fluidity curves of cement paste containing raw fly ash, the addition of raw fly ash improves the fluidity obviously. Compared with cement paste containing 0% fly ash, when 30% raw fly ash is added, the expansion diameter increases by about 20%. Many researches [32,33,34] indicated that the spherical shape and glassy surface of fly ash particles permit greater workability for equal water-binder ratio. The ball-bearing effect of fly ash particles reduces the interface friction, which improves the relative slide among cement particles. Original fly ash shows a noticeable enhancement of workability respect to “only cement” control mortar [28]. Bulk specific gravity has a more decisive influence on fly ash-cement paste viscosity, which differs considerably according to the varieties of fly ash. Different shape of fly ash and cement particles leads to higher bulk specific gravity and lower coefficient of viscosity, which improves the fluidity of cement paste [53]. Meanwhile, spherical particles in cement paste can prevent the aggregation of other particles [54,55].

However, when 40% fly ash added in cement paste, the fluidity tendency appears to be decreasing. It can be concluded that there should be an optimal spherical proportion in cement paste. When too many spherical particles in the mixture, more pores and larger surface area of fly ash particles need more water to help the relative movement of solid particles, which leads to the fluidity decrease of cement paste. Different replacement quantity of fly ash in cement shows different effect on the workability of cement paste [41,42]. In the study of Brown [54], at levels of 10%–40% by volume, for each 10% of fly ash substituted is nearly equal to increase the water content by 3–4%, while further substitution caused rapid decrease in workability.

Fluidity of cement paste containing others fly ash is also shown in Figure 5. The curve of sample VF shows the similar tendency with RF, in which the increasing range is about 16%. From the proportion of spherical destruction in Table 4, there are only 12% spherical particles destroyed and the remainder spherical particles maintain the improvement of fluidity. However, when most of the spherical particles are destroyed (in sample BF1 and BF2 grinding in ball mill), ground fly ash shows little improvement on the fluidity of cement paste, even decrease when BF2 is added. It is proved that the spherical destruction of fly ash particles shows negative effect on the workability of concrete [29,30]. Compared with fly ash grinded by ball mill, sample VF from vertical mill shows some superiority on the fluidity of cement paste. From the particle shape analysis and calculation, more spherical particles in VF show the better fluidity improvement effect than samples BF1 and BF2. Therefore, for different ground fly ash, the decline range is changing with the spherical destruction degree of fly ash particles.

### 4.2. Relation Between Spherical Proportion and Fluidity

The experiment results show the relation between spherical proportion of fly ash and the fluidity of cement paste. The spherical proportion is the proportion in the fly ash-cement system, which can be calculated from the addition and the spherical destruction of fly ash, and the relation curves of different additions are drawn in Figure 6.

The fluidity tendency shown in Figure 6 is similar when the addition of fly ash is different. When fly ash with more undestroyed spherical particles is added in cement paste, the spherical proportion in the system is higher and the fluidity of cement paste improves remarkably. When the addition of fly ash is 30%, the increase range is most obvious. However, when the fly ash addition is large but spherical proportion is small, it is shown that fly ash does not improve the fluidity obviously. It can be observed by the comparison among cement paste samples with the spherical proportion less than 1%. When the most seriously spherical destroyed sample (BF2) is added and the addition is 40%, the fluidity of cement paste is lowest. Therefore, when the effect of the ball-bearing is weaker than the influence of high-water requirement of fly ash, the rheological property of cement paste shows less improvement when fly ash added.

## 5. Strength of Cement Paste Containing Ground Fly Ash

### 5.1. Reaction Degree of Ground Fly Ash

Cement paste samples containing different fly ash (RF, BF1, BF2, VF) are prepared in this experiment, and the cement substitution proportion by fly ash is 50%. The reaction degree of fly ash in different curing periods is tested and listed in Figure 7.

The reaction degree of different fly ash in cement paste is observed in these curves. When raw fly ash is added, the insoluble residue is more than other samples. The reaction degree of raw fly ash is about 5% after 1 d, 21% after 28 d and 33% after 90 d. After grinded in the mill, the reaction rate of fly ash is faster to some extent. More than 36% ground fly ash is consumed in the hydration reaction in 90 days. The improvement difference is more obvious in 21 d, while the reaction degree of BF2 is 40% more than that of raw fly ash. The sequence of reaction degree at less than 28 d age (from high to low) is BF2>BF1>VF>RF, and the difference value between BF2 and VF is about 5% at the age of 14 d and 21 d, which completely agrees to the spherical destruction in Table 4. Meanwhile, the reaction degree of different samples is similar after 28 d, except raw fly ash.

The pozzolanic reaction has a significant impact on the strength of cement paste containing fly ash. Ca(OH)_2_ is produced in the cement hydration process, and OH^-^ ions in pore solution attack on the surface of fly ash [1]. The reaction degree of fly ash is according to the reaction between Ca(OH)_2_ and the glassy particles in fly ash, while the outer layer of these particles are protected by a hard shell, which delay the pozzolanic reaction to a large extent [43]. However, the grinding process destroy the glassy particles, which provide much more surface defects and accelerate the pozzolanic reaction obviously [30,44,45,46,47]. From the SEM images and spherical destruction analysis of fly ash, most glassy particles in fly ash are spherical, and the quantity of spherical particles decreases after grinding process. Therefore, the spherical destruction of fly ash has some certain relation to increment of surface defect and the improvement of reaction degree of fly ash. From the spherical destruction in Table 4, the spherical destruction of VF is much lower than BF1 and BF2, which can explain the phenomenon that, before the age of 28 d, the reaction of VF is much slower than BF1 and BF2.

### 5.2. Compressive Strength of Cement Mortar Containing Ground Fly Ash

Raw fly ash and three different ground fly ash are used to prepare the cement mortar samples. The addition of fly ash is 30%. Cement mortar containing no fly ash (OP) are also prepared as the control sample. The cement mortar samples are curing for 3 d, 7 d, and 28 d, and the compressive strength is tested. The average values of the strength are listed as Figure 8 shown.

Compared to the 100% cement mortar sample, when 30% fly ash added, the compressive strength decreases obviously. From the activity analysis in Figure 7, the reaction degree of fly ash is lower than 30% before the age of 28 d. It is also concluded in plenty of researches [5,21,22,23] that fly ash decrease the early age strength of cement. Meanwhile, the grinding process improves the activity of fly ash and the strength of cement mortar containing ground fly ash is much higher than that containing raw fly ash, especially during early age. From the comparison among different ground fly ash, samples containing BF1, which is grinded for shorter period and shows lower special surface area, shows the lowest compressive strength, while BF2 and VF show similar effect on the strength of cement mortar at the age of 3 d and 7 d. When the curing age is 28 d, the sample containing BF2 shows the highest compressive strength in all cement-fly ash mortar, which is approximately 96% of the sample containing no fly ash.

The influence of fly ash on the strength of cement mortar is mainly from the pozzolanic effect and micro-filling effect [6,7,8,9]. The grinding process leads to the increasement of special surface area and enhances the filling effect of fly ash. When the special surface area of fly ash is grinded to more than 800 m^2^/kg (BF2 and VF), finer particles fill the pore structure in cement mortar more effectively [25] and the strength is much higher than sample BF1 at the age of 7 d and 28 d. The behavior of compressive strength does not always agree with the reaction degree of fly ash. Due to the spherical destruction, BF1 shows higher reaction degree than VF, while mortar containing VF shows much higher compressive strength than that containing BF1. More destroyed non-spherical particles and more surface defects accelerate the reaction of BF1, while the super-fine particles in VF, especially the unreacted fly ash particles, fill the pores and increase the density to enhance the strength of mortar. Therefore, the strength improvement of cement mortar is the combination effect of particle destruction and filling effect by super-fineness particles.

### 5.3. Pore Structure Analysis of Cement Mortar Containing Ground Fly Ash

In order to compare the pore structure of hydration product containing different fly ash, cement paste samples are prepared to analyze the pore distribution. RF, BF2 and VF are used in cement paste and 100% cement paste sample is also prepared. to test the pore structure. The cumulative porosity and incremental intrusion are shown in Figure 9 and Figure 10, Table 5.

From the pore structure comparison in Figure 9, the porosity of cement paste increases obviously after fly ash added. As Table 5 shows, the porosity of sample containing 30% RF is nearly 40% higher than sample containing no fly ash. The reaction speed of fly ash and hydration product is much slower than the hydration speed of cement [5]. Although the unhydrated fly ash particles can fill some pores in cement paste, the porosity in sample containing fly ash is much higher than 100% cement paste. However, ground fly ash shows more effective improvement on the pore structure than raw fly ash. More relatively larger pores (200–5000 nm) can be observed in sample containing RF, while there are more smaller pores (10–100 nm) in sample containing BF2. Compared to raw fly ash, ground fly ash shows higher pozzolanic reactivity and filling effect to transform the larger pores into smaller pores, which improves the compressive strength of cement paste.

Figure 10 shows the pore structure of cement paste containing BF2 and VF, which are exactly similar in the pore size distribution. From the particle size distribution and the spherical destruction analysis, VF shows more super-fineness particles and more undestroyed spherical particles, which can fill more pores in the structure than fly ash. The pore structure analysis agrees to the strength test, which shows that the special surface area and spherical destruction are both the key factors to the pore structure and strength of cement paste.

### 5.4. Micro Morphology of Cement Paste Containing Ground Fly Ash

The micro morphology of cement paste containing different fly ash is also observed by SEM in the experiment. The SEM images of cement paste containing no fly ash, RF, BF2, and VF are shown in Figure 11, Figure 12, Figure 13 and Figure 14.

From the SEM images in Figure 11, the hydration product of cement paste is mainly C–S–H gel and Ca(OH)_2_. When different kinds of fly ash are added into cement paste, spherical fly ash particles can be observed in the images, especially RF and VF. Figure 12 shows that the reaction between raw fly ash and Ca(OH)_2_ is slow and more pores are observed in the hardened cement paste. In comparison, with VF, as Figure 14 shows, the particle shape is similar to RF and the particle size is smaller, while the surrounding area of fly ash particles is much denser than that shown in Figure 12. Fly ash in Figure 13 is the ground fly ash milled in the ball mill and the spherical shape is destroyed seriously. Less spherical particles are observed in the image, which shows that the non-spherical particles more easily react with the hydration product to improve the pore structure in cement paste and enhance the compressive strength.

## 6. Conclusions

The special surface area, particle size distribution, and particle shape of ground fly ash are analyzed, and the spherical distribution of different ground fly ash is calculated on the basis of SEM images. The fluidity and compressive strength of cement paste are tested, while the reaction degree of fly ash, the pore structure and SEM images of cement paste are analyzed to study the mechanism. The relation of spherical destruction and fluidity, strength is studied in this article, and the following conclusions can be drawn:
Fly ash ground in different mill systems show different particle sizes and morphology. Ground fly ash by vertical mill with separation system shows finer particle size distribution, while that by ball mill shows higher spherical destruction. From the analysis of SEM images of ground fly ash particles, more than 80% spherical particles of fly ash are destroyed, becoming non-spherical particles, after grinding in a ball mill, while the spherical destruction in the vertical mill grinding process is only 11.9%.Spherical particles in fly ash-cement system improve the fluidity of cement paste, and the improvement effect is remarkably weaker after grinding process. Spherical destruction shows a significant relation with the ball-bearing effect and the fluidity. A higher spherical proportion leads to higher fluidity and the increase in range is more obvious when the addition of fly ash is 30%.Particle size distribution and spherical destruction are both the key factors to the activity of ground fly ash and the strength of cement paste. The ball mill destroys the spherical particles and accelerates the pozzolanic reaction, while the vertical mill produces the super-fine particles and the filling effect also enhances the strength of cement mortar.Spherical destruction is an important measurement index to evaluate the grinding effect of the fly ash mill. The calculation on the basis of SEM images is an effective method to measure the spherical proportion in ground fly ash.


## Figures and Tables

**Figure 1 materials-14-00283-f001:**
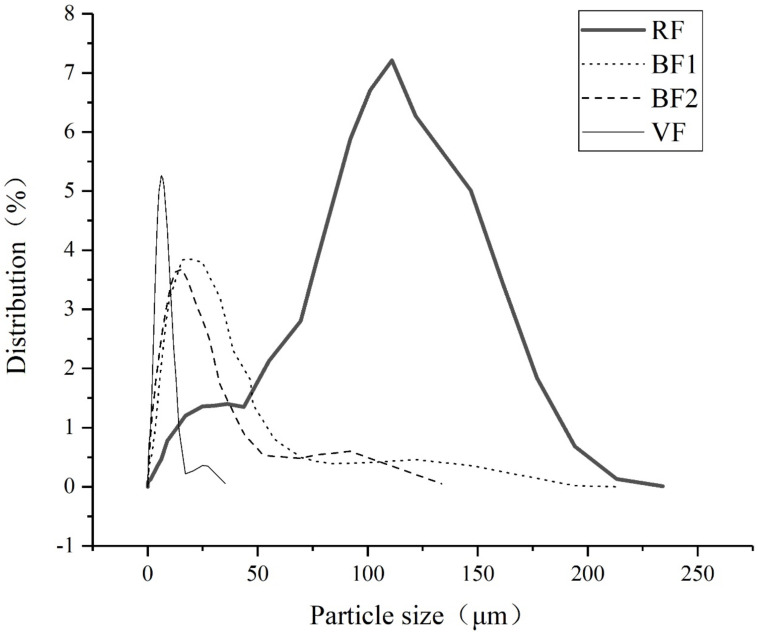
Particle size distribution of different fly ash.

**Figure 2 materials-14-00283-f002:**
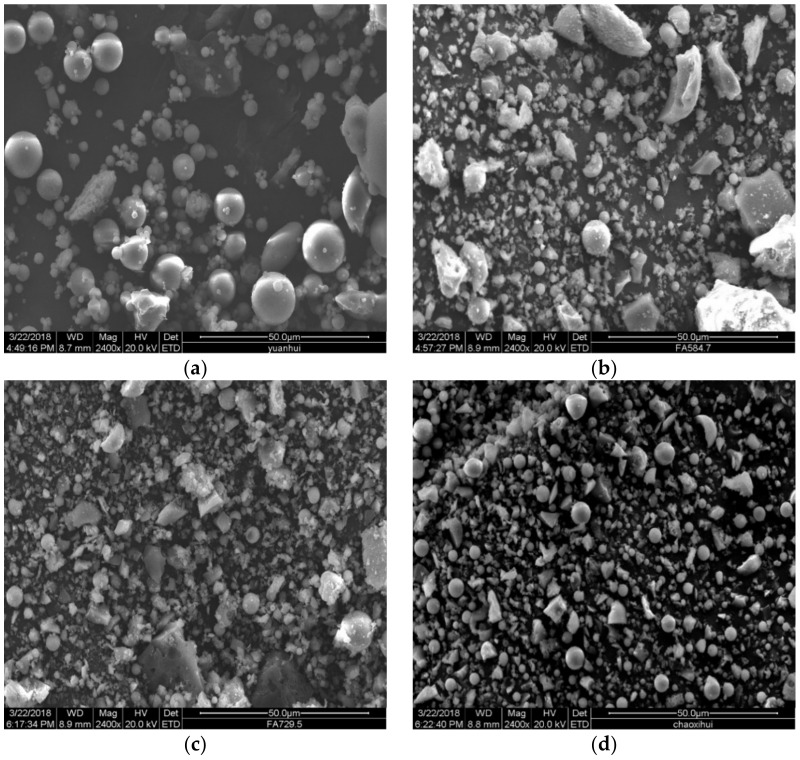
SEM images of different fly ash. (**a**) Raw fly ash (RF). (**b**) Grinding 1h in ball mill (BF1). (**c**) Grinding 2h in ball mill (BF2). (**d**) Grinding in vertical mill (VF).

**Figure 3 materials-14-00283-f003:**
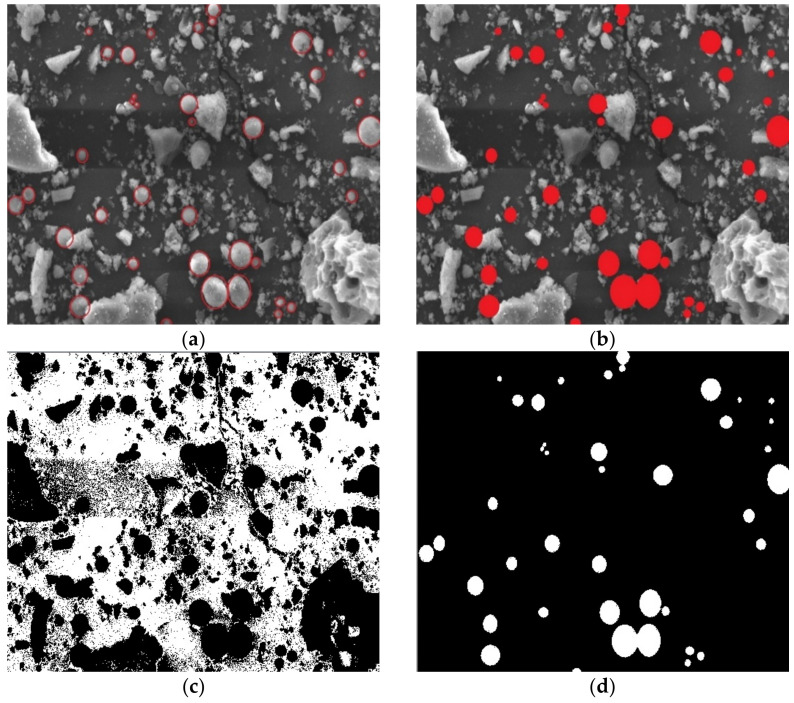
Area calculation in SEM images. (**a**) Spherical particles. (**b**) Coloring of spherical particles. (**c**) Interspace area. (**d**) Spherical particles area.

**Figure 4 materials-14-00283-f004:**
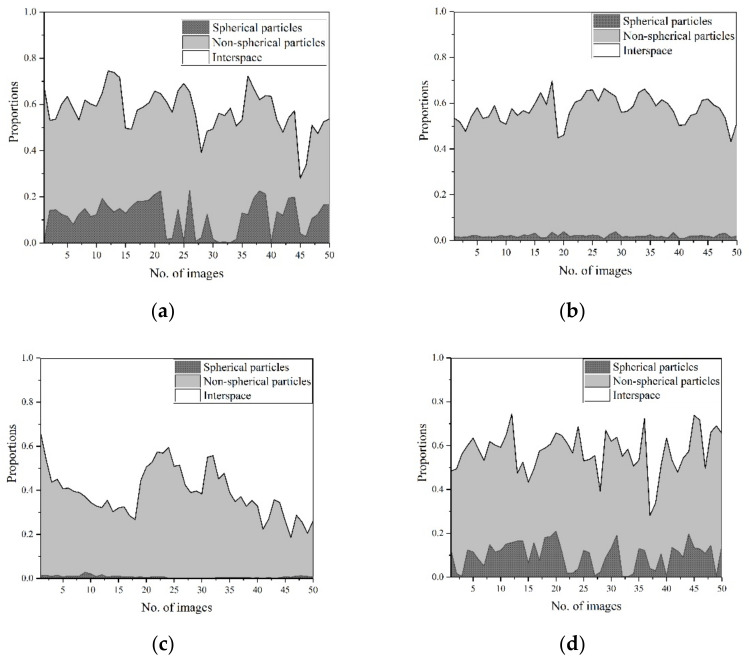
Accumulating area of different fly ash. (**a**) RF (Raw fly ash). (**b**) BF1 (Grinding for 1 h in ball mill). (**c**) BF1 (Grinding for 2 h in ball mill). (**d**) VF (Grinding in vertical mill).

**Figure 5 materials-14-00283-f005:**
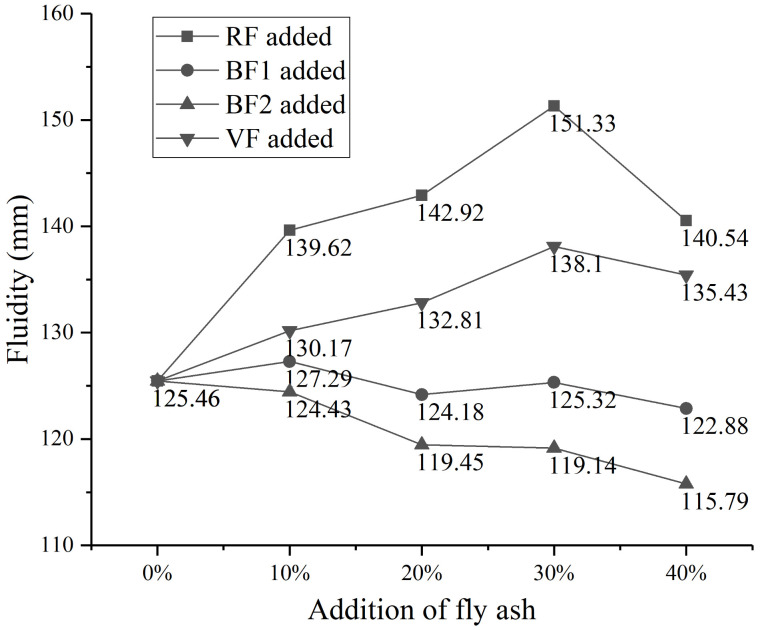
Fluidity of cement paste containing different fly ash.

**Figure 6 materials-14-00283-f006:**
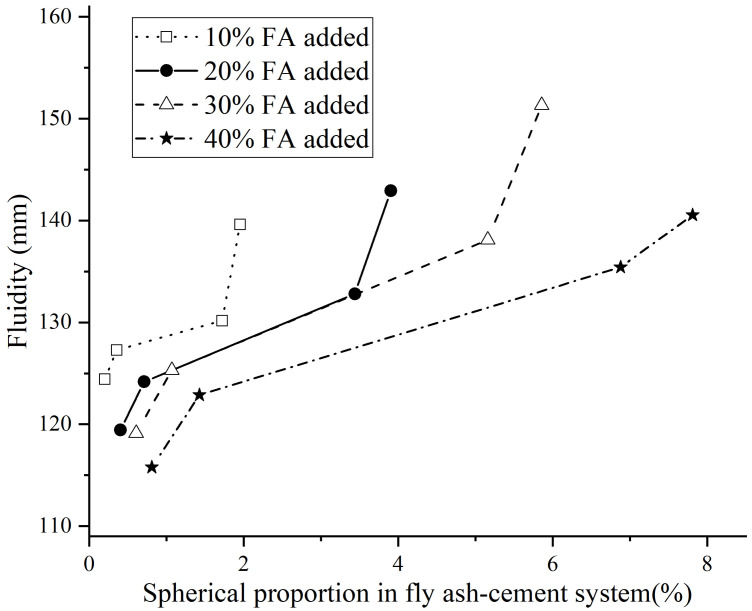
Relation of fluidity and spherical proportion (W/B = 0.5).

**Figure 7 materials-14-00283-f007:**
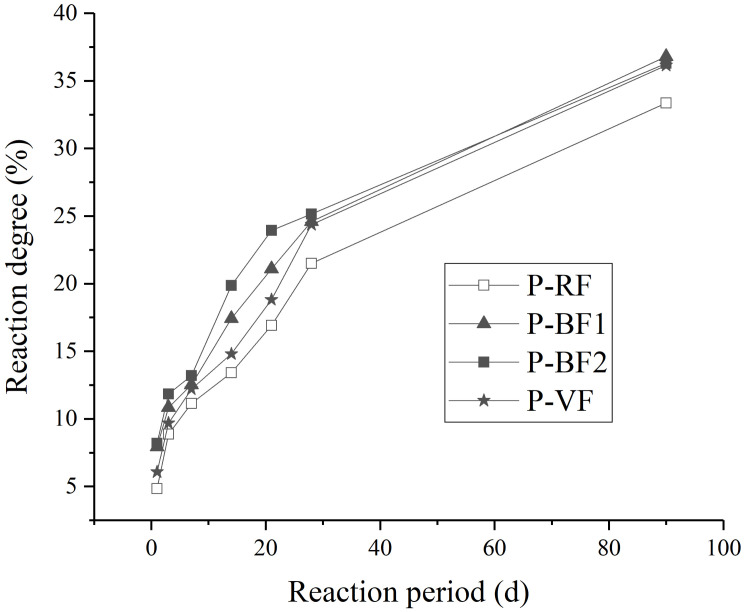
Reaction degree in different curing periods.

**Figure 8 materials-14-00283-f008:**
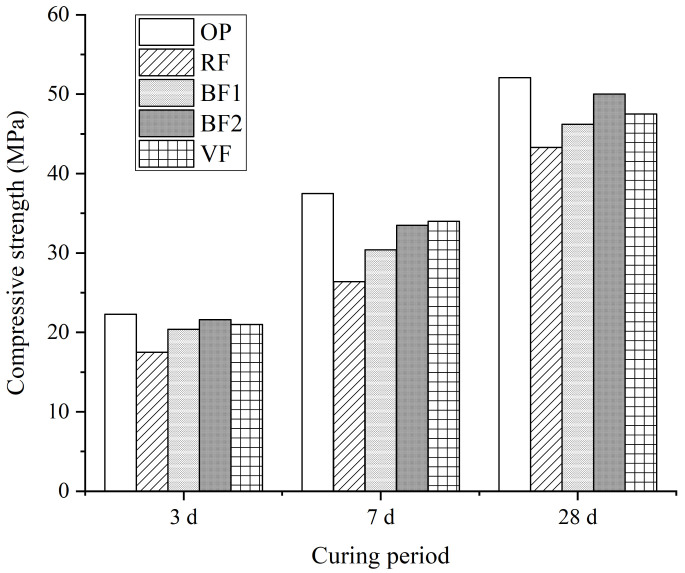
Compressive strength of cement mortar containing ground fly ash.

**Figure 9 materials-14-00283-f009:**
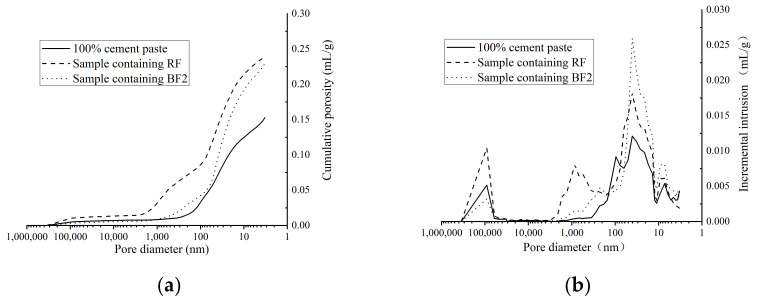
Pore structure of samples containing no, raw and ground fly ash. (**a**) Cumulative porosity. (**b**) Incremental intrusion.

**Figure 10 materials-14-00283-f010:**
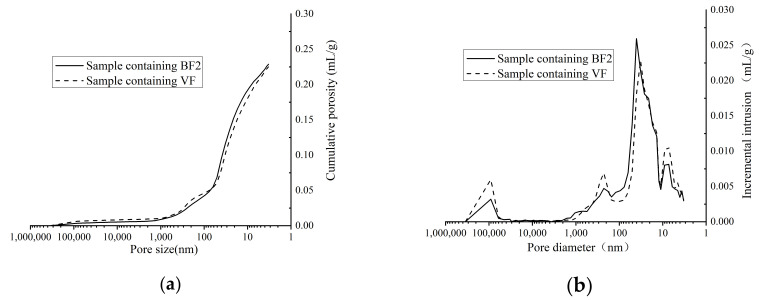
Pore structure of samples containing ground fly ash in different mill. (**a**) Cumulative porosity. (**b**) Incremental intrusion.

**Figure 11 materials-14-00283-f011:**
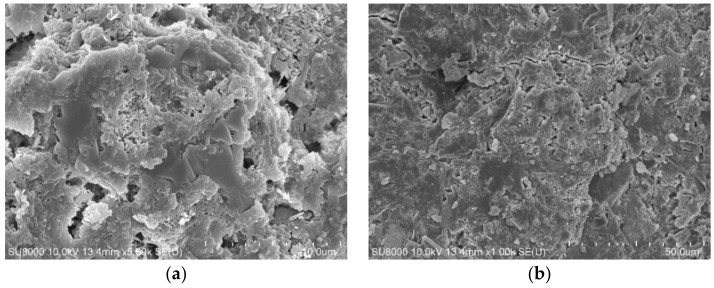
SEM images of 100% cement paste. (**a**) ×5000. (**b**) ×1000.

**Figure 12 materials-14-00283-f012:**
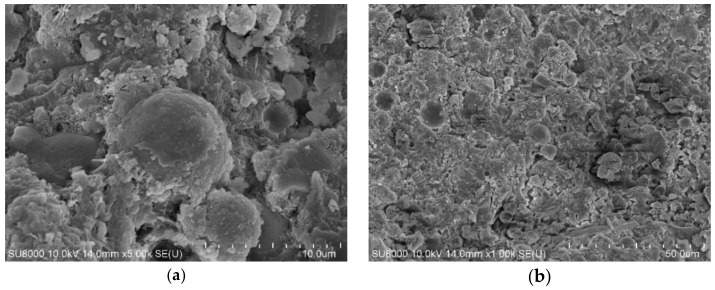
SEM images of sample containing RF. (**a**) ×5000. (**b**) ×1000.

**Figure 13 materials-14-00283-f013:**
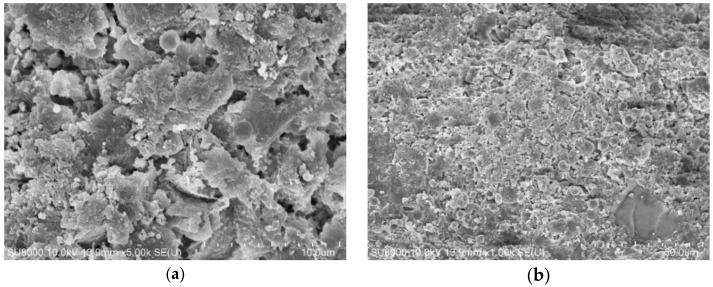
SEM images of sample containing BF2. (**a**) ×5000. (**b**) ×1000.

**Figure 14 materials-14-00283-f014:**
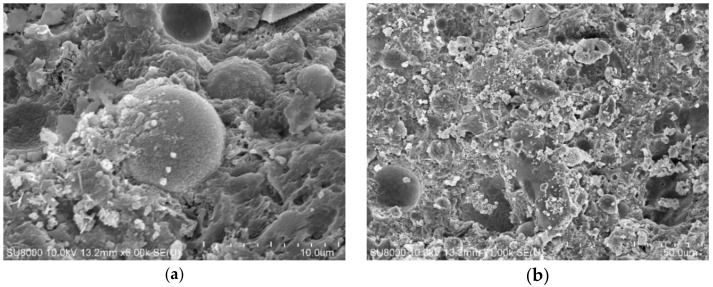
SEM images of sample containing VF. (**a**) ×5000. (**b**) ×1000.

**Table 1 materials-14-00283-t001:** Chemical composition of fly ash and cement (by weight, %).

Samples	SiO_2_	Al_2_O_3_	Fe_2_O_3_	CaO	MgO	SO_3_	Na_2_O	Loss
Cement	20.63	4.45	2.88	64.06	1.67	2.88	0.54	1.55
Fly Ash	53.72	28.11	11.55	3.54	0.78	0.42	0.75	0.98

**Table 2 materials-14-00283-t002:** Different grinding conditions.

No.	Mill Type	Grinding Condition
RF	–	Raw fly ash
BF-1	Ball mill	Grinded for 1 h
BF-2	Ball mill	Grinded for 2 h
VF	Vertical mill	Grinded in grinding system with separation system

**Table 3 materials-14-00283-t003:** Special surface area and average particle size of fly ash.

Samples	RF	BF1	BF2	VF
Special Surface Area/m^2^·kg^−1^	410.6	666.9	803.0	897.6
Average Particle Size/μm	53.94	10.16	9.16	6.27

**Table 4 materials-14-00283-t004:** Spherical destruction in different fly ash.

Samples	RF	BF1	BF2	VF
Spherical Proportion	19.53%	3.57%	2.03%	17.20%
Spherical Destruction	0%	81.7%	89.6%	11.9%

**Table 5 materials-14-00283-t005:** Porosity of different cement paste.

Sample	Porosity
100% Cement Paste	22.4%
Sample Containing 30% RF	31.1%
Sample Containing 30% BF2	30.5%
Sample Containing 30% VF	30.1%

## Data Availability

Data available in a publicly accessible repository.

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
