# Peer review of "Influence of Particle Morphology of Ground Fly Ash on the Fluidity and Strength of Cement Paste"

_materials, 2021, doi:10.3390/ma14020283_

Round 1

Reviewer 1 Report

Some comments:

1)  Please check equation (1). In the reviewer opinion, the formula may use W/B instead of rw/b.

2)  We only use the surname of the first author to cite the reference in the text (line 53 and 159)

3)  In Figure 8 as a reference is treated RF cement paste not paste without fly ash.

Author Response

Thank you for your hard work and recognition of this paper. Those comments are very helpful for revising and improving our paper. We have studied the comments carefully and made corrections which we hope meet with approval. The main corrections are in the manuscript and the responds to the reviewers’ comments are as follows.

1) Please check equation (1). In the reviewer opinion, the formula may use W/B instead of rw/b.

Reply:
We have revised rw/b to w/b in the formula. The changes have been highlighted in red color.

2) We only use the surname of the first author to cite the reference in the text (line 53 and 159)

Reply:
We have revised the name of the author. The changes have been highlighted in red color.

3) In Figure 8 as a reference is treated RF cement paste not paste without fly ash.

Reply:
The comparison is aimed to study the effect of particle size and morphology of fly ash on the strength. In the figure, we use RF cement paste as the reference. We have revised "Reference sample" to "OP" in the figure, and the changes in the paper have been highlighted in red color.

Reviewer 2 Report

Row 32: Fly ash is one of the first artificial admixture..... is not corect because Types of Concrete Admixtures are:
  • air entrainers.
  • water reducers.
  • set retarders.
  • set accelerators.
  • superplasticizers.
  • specialty admixtures: which include corrosion inhibitors, shrinkage control, alkali-silica reactivity inhibitors, and coloring.

The corect term for fly ash is addition 

because

EN-206-1 defines additions as finely divided materials used in concrete in order to improve certain properties or to achieved special properties. This standard deals with two types of inorganic additions:
- type I - nearly inert additions,
- type II – pozzolanic or latent hydraulic additions.
General suitability as type I addition is established for:
- filler aggregate conforming to prEN 12620:1996,
- pigments conforming to EN 12878.
General suitability as type II addition is established
for:
- fly ash conforming of EN 450,
- silica fume conforming to prEN 13263: 1998.

Author Response

Thank you for your hard work and recognition of this paper. We have studied the difference of addition and admixture. Fly ash can be called "addition" or "mineral admixture". We have revised "admixture" to "addition" in the first paragraph, and the changes have been highlighted in red color.

Reviewer 3 Report

the topic of the article falls within the scope of the journal. please review the English, it is generally difficult to read. I attach a pdf with some comments. Please in case of resubmission provide your comments/responses on this same pdf.

Author Response

Thank you for your hard work and recognition of this paper.Those comments are very helpful for revising and improving our paper. We have studied the comments carefully and made corrections which we hope meet with approval. The main corrections are in the manuscript and the responds to the reviewers’ comments are as follows. Meanwhile, the responses are also submitted in the same pdf.

1) contact info missing, emails phones

Reply: The email is added in the paper and highlighted in red color;

2) for your particular setup which we do not know yet. try to redraft this part so it is clear by itself

Reply: To make it much clearer, this sentence is revised to " To different addition of ground fly ash, the fluidity of cement paste decreases with the increase of spherical destruction obviously." The changes have been highlighted in red color.

3) please check the manuscript. I will not review the writing. I do not know how many times you wrote fly ash in just one paragraph. Please read the whole text again and check for errors, flow while reading, etc.

Reply: We use too many times "fly ash" in this paragraph. To improve the paragraph, some sentence is revised. The fourth sentence of the paragraph is revised to "In the previous studies of fly ash, the pozzolanic effect [6-8] and filling effect [9] improve the strength of concrete, while the ball-bearing effect [10, 11] enhances the [12]fluidity of concrete." and highlighted in red color. The others errors in the manuscript are also revised and highlighted in red color.

4) how are defined these categories?

Reply: In China, fly ash is classified into three grades "Ⅰ, Ⅱ, Ⅲ" by the fineness. Class I is the finest. It is added in the paper and highlighted in red color.

5) Reply: "a great many of" is not proper here. We have revised it to "a lot of" and highlight it in red color.

6) Reply: "r/min" is revised to "rpm" and highlighted in red color.

7) Reply: The unit "(%)" is added in the table.

8) Reply: the medium particle size (d50) means the particle size of 50% particles is above this value while 50% below this value. It is an important factor to describe the size distribution. We also list the average particle size (the value of d50) in the table 3. Figure 1 and Table 3 are both used to describe the size distribution. Therefore, Figure 1 is to list the distribution curve, and the value in Table 3 is much clearer to quantify the special surface area and the particle size. So, it is also neccesary to the analysis.
As mentioned above, to the Figure 1, the curves are only show the distribution of particle size, which is difficult to quantify it. Therefore, the quantity of particles between 50μm and 200μm can only be described as "most" here.

9) Reply: To the quality of the figure, we will submit all the original figure to the editor.

10) average of 3 specimens from the same group or average of different groups? please check the whole text, sometimes is not clear what you really mean.

Reply: We have prepared three groups of specimens to test the 3d, 7d and 28d strength. To every age, 3 specimens are chosen from different groups to avoid the measuring error as much as possible. For lack of space, we have not listed it in the paper.

11) it would be good if you can upload them as complementary material,

Reply: We will submit to the editor. Thank you for your suggestion.

12) since you have many measurements it could be good to include also the std dev on the plots

Reply: The suggestion is very useful. We are trying to do it in the future study.

13) same as previous figure

Reply: Figure 5 and Figure 6 show the same data. Therefore, Figure 6 shows the relation of fluidity and spherical proportion to different fly ash addition. It is meanful to the analysis.

14) Reply: "Reference samples" is not proper here. Another reviewer also mentioned it. We have drawn this figure again and renamed it. In the line 322, "reference sample" is revised to a clearer expression. The changes have been highlighted in red color.

15) use the same format for all references

Reply: We have used the same format for all references, except the first one because it is a book, not an article. Thank you for your suggestion.

Round 2

Reviewer 3 Report

Some comments from the first review have not been replied. I add a pdf again with some old and new comments

Author Response

Thank you for your comments again. We apologize for some mistakes in the first reply. We have studied the comments again and made corrections. Thank you for your patient explanation. The main corrections are in the manuscript and the responds to the comments are as follows:

1) Reply: "obviously" is not proper here. We have delete this word and highlighted it in the yellow background.

2) Reply: The classification of fly ash is confirmed to the Chinese National Standard GB/T 1596-2017. We have added it in the paper and highlighted it in the yellow background.

3) Reply: The chemical composition of fly ash and cement is by weight. We have added it in the paper and highlighted it in the yellow background.

4) Reply: Thank you for your patient explanation. We forgot to list the test method at first. Fly ash is dispersed at first and analyzed by LS13320 laser particle size analyzer. We have added it in the paper and highlighted it in the yellow background.

5) Reply: We have calculate the particles distribution in the range of 50μm to 200μm from the raw data. About 82% of particles are concentrated between 50μm and 200μm. We have added it in the paper and highlighted it in the yellow background.

6) Reply: We apologize for the mistake in describing the curing condition and testing method. The curing temperature, relative humidity and the loading speed of compressive test are added in the paper and highlighted in the yellow background.

7) Reply: Thank you for your comments about the fluidity test. In Figure 5 we use four different addition of different fly ash. We have found that the fluidity improves with the increase of spherical proportion. Of course if there are more data (the addition of fly ash, fly ash with different spherical proportion), the law will be more clear. However, we apologize for the lack of the same batch of fly ash. We will take more experiment in the future work to explore the law and mechanism.

8) Reply: Although we have not understand your comments in Figure 8, we think the comparison between RF and BF2 is not proper in this Figure. Therefore, we delete the comparison here. Figure 8 is changed here and the original figure is submitted with the revised manuscript.

Thank you for your comments again.